# Jan van Eyck's *New York Diptych*: A New Reading on the Skeleton of the Great Chasm

## Miyako Sugiyama

Japan Society for the Promotion of Science, Tokyo 102-0083, Japan; miyakosugiyama1129@gmail.com

**Abstract:** The *Crucifixion and Last Judgment*, or the so-called *New York Diptych*, is one of the most controversial paintings attributed to Jan van Eyck (ca. 1390–1441) and his workshop. For well over a century, art historians have vigorously discussed its attribution, composition, functional intent, and even its dating. In light of prior scholarship addressing these remarkable panels, this paper focuses on the skeleton represented in the *Last Judgment* to reveal its iconographical meanings. Specifically, I highlight the inscriptions written on the skeleton's wings, suggesting that the texts were cited from an All Saints' Day sermon delivered by the Burgundian abbot Bernard of Clairvaux (1090–1153) who discussed a temporal location for blessed or sinful souls.

**Keywords:** Jan van Eyck; the *New York Diptych*; Bernard of Clairvaux

## 1. Introduction

The two panels of the *New York Diptych*[1] (Figure 1) depict two very different scenes in the story of Christ: his crucifixion in Jerusalem and the Last Judgment, the latter featuring a gruesome depiction of hell presided over by a winged skeleton, on which we read "CHAOS MAGNVM" and "VMBRA MORTIS". What prompted this analysis is that the inscriptions on the wings have received little attention and their genesis and meaning have been overlooked. The online catalogue for the Metropolitan Museum of Art in New York does not indicate the source of the inscriptions, but as Hans Belting and Dagmar Eichberger point out, "CHAOS MAGN3VM" is mentioned in several places in the Bible (Luke 1:79, Psalms 87:7, Job3:5, 10, 22), while "VMBRA MORTIS" is mentioned only in Luke 16:26 (Eichberger 1987, p. 87, nr. 488). There is no biblical text, however, in which both phrases are mentioned together. One can only make assumptions as to why Van Eyck included the two inscriptions on the skeleton's wings, but as Belting and Eichberger surmise, they are likely metaphors for the underworld. After considering previous discussions of the panels, this paper proposes a plausible, more specific source of these texts. As detailed in the following sections, the current author suggests that the inclusion of these phrases was inspired by a sermon delivered by Bernard of Clairvaux, who argued for an inescapable destiny for the sinful.

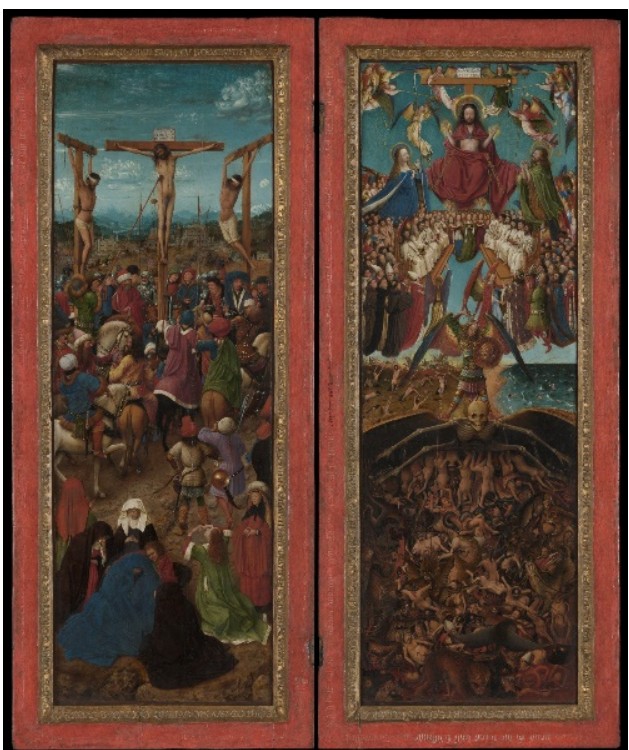

**Figure 1.** Jan van Eyck and his workshop, *Crucifixion and Last Judgment* (*the New York Diptych*), ca. 1440–1441, New York, Metropolitan Museum of Art.

## 2. Overview of the Work

On the left panel of the *New York Diptych*, Christ is nailed to a cross, his chest stabbed with the Lance of Longinus. He is surrounded by Jewish priests, soldiers, and noblemen on horses, most bearing distorted, ugly smiles likely to imply their sinful state (Figure 2). On the top of the cross, we read "IHC · NAZAR[ENVS] · REX · IVDE[ORVM]" with its Greek and Hebrew translations (Christiansen and Ainsworth 1999, p. 86). In the foreground is the Virgin Mary, covered almost completely with a blue cloak; inconsolable, she is attended by John the Evangelist (Figure 3). Mary Magdalena is also depicted in the panel; she is prostrate and staring up at the cross, her fingers interlaced and positioned in front of her face. The woman next to her who is wearing an elaborate turban is generally considered to be the Erythraean or Cumaean Sibyl. At her opposite side is another woman with a turban, who is also considered to be of the sibyls (Panofsky 1953, p. 240, n. 1; Eichberger 1987, p. 80).

Shifting to the right panel, the heavenly Christ, who overcame death on the cross, now wears a red mantle and is positioned to give justice on Judgment Day (Figure 4). Situated to the left is the Virgin of Mercy wearing an expansive blue mantle enveloping the good souls she protects. Similarly, John the Baptist is cloaked in green, his saintly halo indicating his righteousness. Below them are the twelve apostles sitting on choir stalls, on the left side of which is a relief of the Original Sin (Figure 5), symbolizing the redemption that will be given to some as a result of Christ's death on the cross. Around the apostles are holy virgins, church authorities, and noblemen. These groups of holy figures compose the *Allerheiligenbild* (All-Saints image), which is comparable to the composition of the *Adoration of the Lamb* by Hubert and Jan van Eyck (Figure 6).

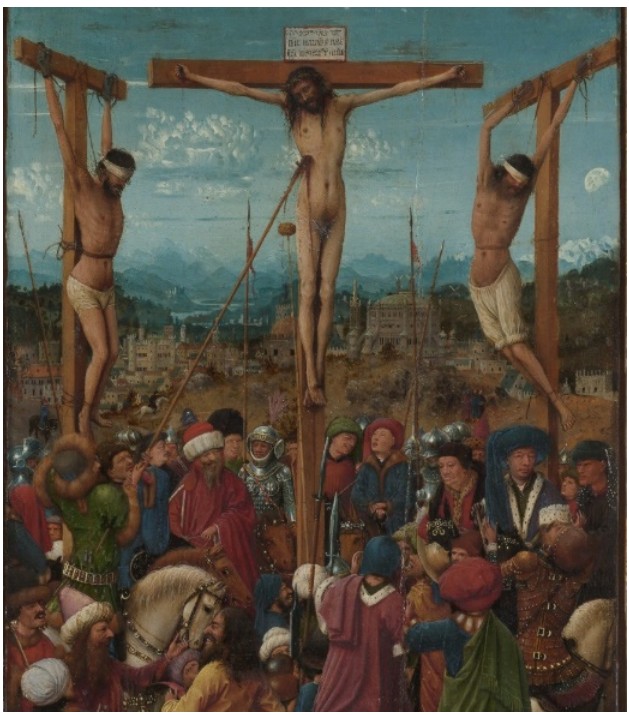

**Figure 2.** Detail of Figure 1.

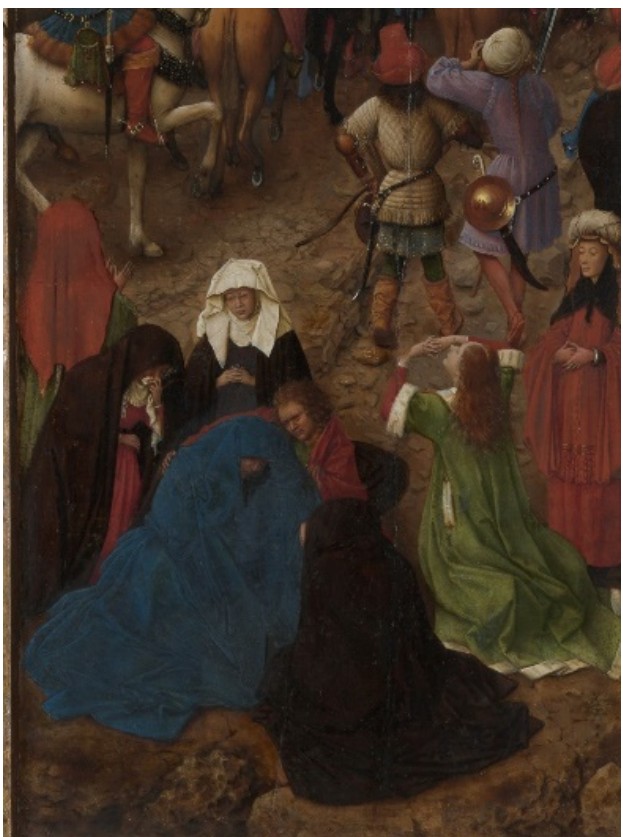

**Figure 3.** Detail of Figure 1.

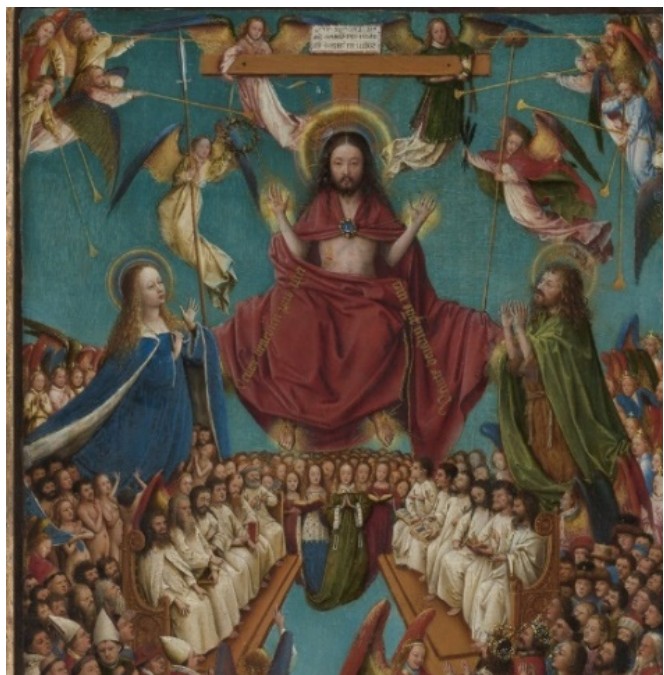

**Figure 4.** Detail of Figure 1.

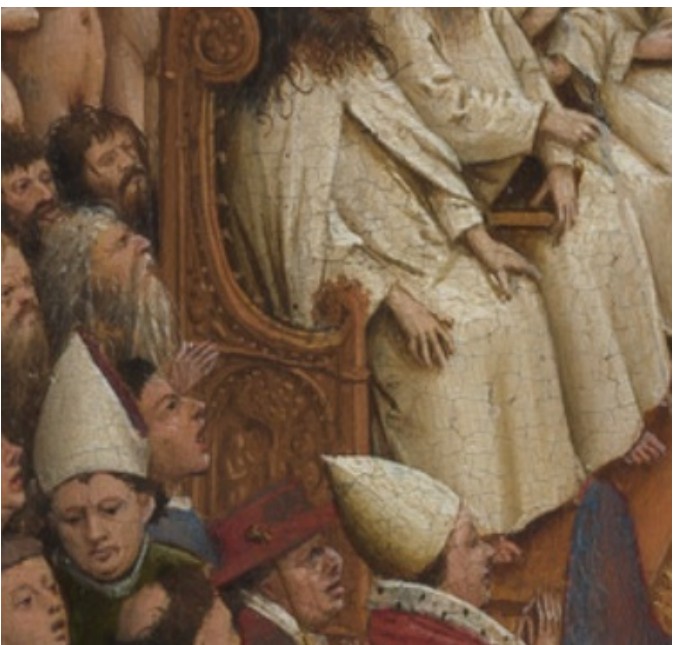

**Figure 5.** Detail of Figure 1.

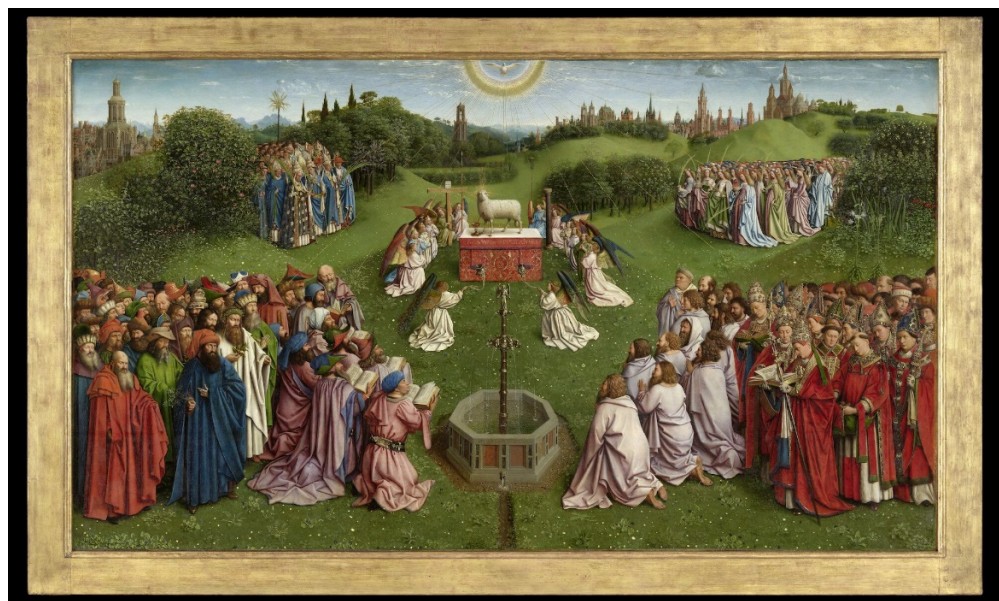

**Figure 6.** Hubert and Jan van Eyck, *The Adoration of the Lamb*, central panel of the *Ghent Altarpiece*, 1432, Ghent, St. Bavo Cathedral.

Under the merciful vision of heaven is a land devastated by earthquake and fire (Figure 7). To be judged, souls are resurrected from their earthly tombs, while others cross a stormy sea. On the boundary between heaven and hell, as well as land and sea, the winged Archangel Michael awaits. While traditionally depicted as holding a scale to weigh the good and the evil during the Last Judgment, as can be seen in the *Beaune Altarpiece* by Rogier van der Weyden, here he wears colorful armor and holds a sword and shield. At Michael's feet is a skeleton that stretches out its limbs and wings. From its abdominal area, sinful souls, including bishops, noblemen, and a pregnant woman, are ejected to be tortured by demons in hell.

Latin inscriptions written on the painted surface and frames of the *New York Diptych* are positioned to correspond with the juxtaposed representations. On the frame of the Crucifixion, the Latin texts read:

DOMINVS POSVIT I EO INIQVITATE OMNIVM NRM OBLATVS E QVIA IPE VOLVIT + NON APERVIT OS SVV SIC OVIS AD OCCASIONE DVCET + QI[?] AGNVS CORAM TONDETE SE OBMVTESCET PPT SCEL PPLI MEI PERPER-CVSSI EV ET DABIT IMPIOS PRO SEPVLTVRA ET DICITES PRO MORTE SVA VSAE[?] [T]R[ADI]DT I MORTE AIAM SVA + CV SCELERATIS REPV[T]ATVS E ET IPE PCCM MVLTORV TVLIT + PRO TRANSGRESSORIBVS ORAVIT. (And the Lord hath laid on him the iniquity of us all. He was oppressed, and he was afflicted, yet he opened not his mouth: he is brought as a lamb to the slaughter, and as a sheep before her shearers is dumb, so he openeth not his mouth . . . For the transgression of my people was he stricken. And he made his grave with the wicked, and with the rich in his death . . . and he was numbered with the transgressors; and he bare the sin of many, and made intercession for the transgressors [Isaiah 53:6–9,12].) (Christiansen and Ainsworth 1999, pp. 86, 88).

Additional biblical texts frame the Last Judgment panel:

DED MORS MORTVOS ECCE TABNACLM DEI CV HOIBVS + HITAB CV EIS IPI PP EI ERVNT + IPE DS CV EIS EIT EOR[D]S ET ABSTG OEM LA[G]A AB OCLIS SCOR + MORS VLT NON EIT N[Q]LVC NQDOLOR EIT VLTRA DEDIT MARE MORTVOS SVO CONGREGABO SR EOS MALA SAGITTAS MEA OPLEBO I EIS OSVET FAME + DEVORABT EO AVE MORSV A[?] DESTES BESTIAR MITTA I EOS CV FORE THECIV SR T[?]A AQ SPECIV. (And death and hell delivered

up the dead which were in them [Revelation 20:13]; Behold, the tabernacle of God is with men, and he will dwell with them, and they shall be his people, and God himself shall be with them, and be their God. And God shall wipe away all tears from their eyes; and there shall be more death, neither sorrow, nor crying, neither shall there be any more pain [Revelation 21:3,4]; And the sea gave up the dead which were in it [Revelation 20:13]; I will heap mischiefs upon them; I will spend mine arrows upon them. They shall be burnt with hunger, and devoured with burning heat, and with bitter destruction: I will also send the teeth of beasts upon them, with the poison of serpents of the dust [Deuteronomy 32:23,24].) (Christiansen and Ainsworth 1999, p. 88).

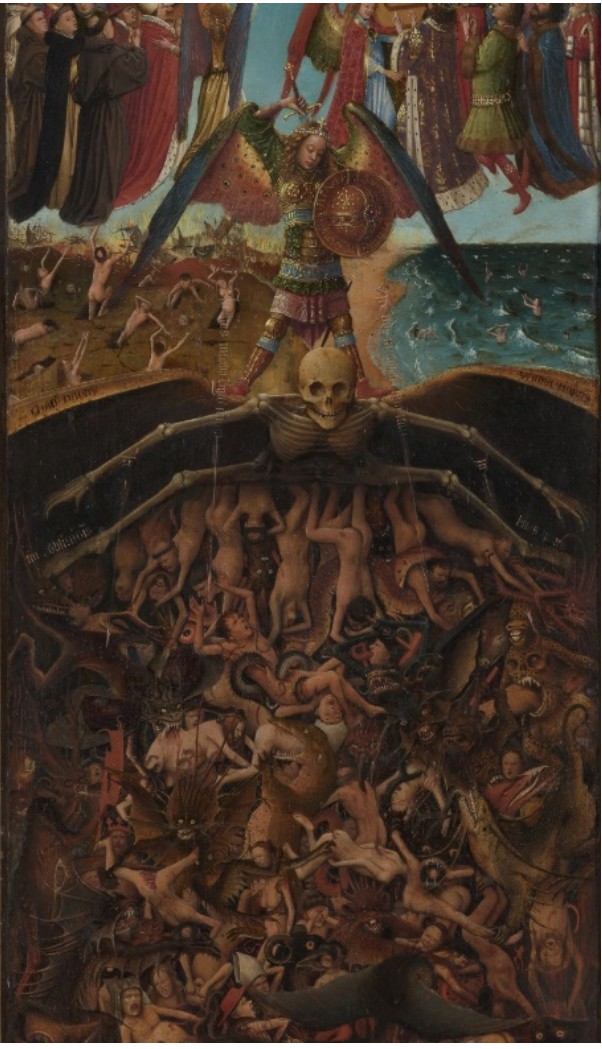

**Figure 7.** Detail of Figure 1.

The inscriptions on the frames are written with a pastiglia technique to provide a low relief decoration in gesso, which is distinctive to Van Eyck's work. Although scholars have debated whether the inscriptions are original to the panels, recent scientific examination has confirmed that they were, indeed, painted at the same time as the images and that a Flemish translation of a part of the Latin text is written in the Gothic style on the lower frame (Ainsworth 2017, p. 230). Other inscriptions are written within the Last Judgment. On both sides of the red mantle of Christ, we read "Venite benedi[c]ti p[at]ris mei (Come, ye blessed of my Father)" [Matthew 25:34] (Christiansen and Ainsworth 1999, p. 86). Similarly, to the right and the left of Michael are the words, " . . . vos maledi[ct]I i[n] ignem [aeternum?] ( . . . ye cursed, into everlasting fire)" [Matthew 25:41] (Christiansen and Ainsworth 1999,

p. 86). The edge of the text is transformed into an arrowhead, which alludes to the text on the frame: "I will spend mine arrows upon them". Although it is difficult to read the text on Michael's armor and shield, Eichberger suggests that they are Greek translations of VINDE(X) IUST(ITIA) J(ESU) CH(RISTI) (avenger of justice. Jesus Christ) and AGLA (a mystical sign)-ADONAY (one of the Biblical Hebrew names of God)-TETRAGRAMMATON (the four-letter Hebrew word translated as YHWH, Yahweh) (Eichberger 1987, p. 87).

The two panels, framed and accompanied with the texts, represent and—literally—tell the stories of Christ's salvation on the earth and in heaven. According to Erwin Panofsky, the *New York Diptych* represents both an earthly city and the heavenly city (the New Jerusalem) as described by Augustine of Hippo in his fifth-century *City of God* (Panofsky 1953, p. 238). While the heavenly city is hidden within the earthly city, when the final judgment day comes both heaven and hell will be revealed. Augustine cites "Then death and Hades were thrown into the lake of fire" (Revelation 20:14) and mentions that death and Hades are demons. In the *New York Diptych*, Death and Hades are combined in the fearsome winged skeleton, on which we read CHAOS MAGNV[M] (Great Chasm)/VMBRA MORTIS (Shadow of Death) (Figure 8). Below the skeleton's legs are illegible texts. Belting and Eichberger suggest "Terra oblivion (?) lacus" (Belting and Eichberger 1983, p. 106), while Jacques Paviot assumes the text under the left leg is "me. Obliuisc(er)is"; however, the text on the right is impossible to read (Paviot 2010, 73. Bd., H. 2, p. 159).

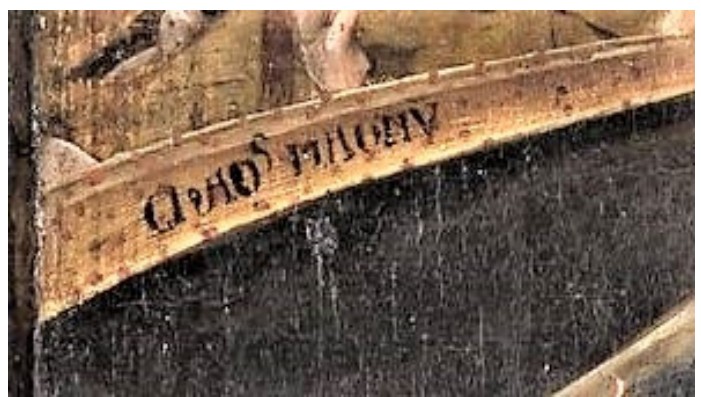
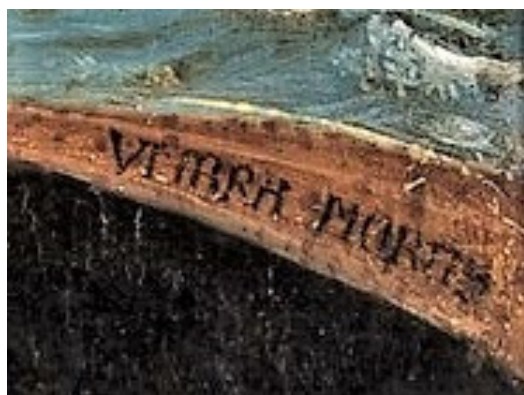

**Figure 8.** Detail of Figure 1.

### 2.1. Original Configuration and Provenance

The subject matter of the *New York Diptych*, which depicts Christ's death on earth and rebirth in heavenly Jerusalem, reminds us of his sacrifice and victory over death. Both the Crucifixion and Last Judgment are subjects that have often been represented in the central panel of a triptych, an altarpiece consisting of three panels to be placed above or behind the altar. A triptych with a fixed central panel and movable wings was common in the Netherlands. It can be opened and closed, allowing various representations to be introduced to the audience. Iconographies related to Christ's salvation were often chosen to be depicted in the central panel (Lane 1984; Humfrey and Kemp 1990). Indeed, it would be difficult to identify another subject more important than the Crucifixion and Last Judgment in the middle panel. While the larger meaning of the diptych is not in question, its provenance tells a different story.

According to Johann Passavant, the piece was once in the collection of a Russian diplomat, Dmitry Tatischev, who purchased the panels from a convent in Spain in 1841 (Passavant 1841, p. 9). Reportedly, at that time the two panels we see today were attached to a larger panel that showed the Adoration of the Magi, and thus originally composed a triptych. The Adoration of the Magi seems to have been lost at a certain point; moreover, Passavant indicated that the reverse sides of the two panels featured two saints in grisaille, which have been lost to time as well. On the basis of his report, it had long been considered

that the panels once existed in Spain in the fifteenth century, possibly in the Castilian region (Brans 1952, p. 41).

A recent discovery by Gabriele Finaldi revealed that the panels could be found in Naples in the mid-seventeenth century. According to a new document, the two panels we see today were in the collection of Ramiro Núñez Felípez de Guzmán, Duke of Medina de las Torres and Spanish Viceroy of Naples (c. 1600–1668), and his second wife Anna Carafa della Stadera, Principessa of Stigliano (1607–1664) in 1641:

> Dui quadretti lungho attaccati insieme con tre colonne con base e capitelli et architrave sopratutto indorato con dio Padre nella sumita et uno di detti quadri vi e la Crocefissione di Nostro Sigre in mezzo li dui ladroni co'diuerse figurine di Soltadi, e le Marie con lettere antiche intirno, nell'altro Giesu Christo appoggiatto alla Croce con diverse figure d'Angeli e sotto esso infinite figure di santi, sotto a'quali il Purgatorio co'un Angelo in mezzo e sotto l'inferno co'lettere similmente intorno di mano del detto Alberto dura (Two long squares attached with three columns with a base and capitals as well as an architrave with gilded God the Father. One of the panels shows the Crucifixion of Our Savior in the middle accompanied with two thieves, several soldiers, and the Virgin Mary. It is surrounded with ancient letters. The other panel shows Christ against the cross and various items held by angels. Below him are countless saints; below them is the purgatory with an angel in the center; and under hell we find letters similar to those mentioned above. By the hands of Albrecht Dürer.) (Jones 2014, p. 37. Translation by the current author).

Although attributed to Albrecht Dürer, the panels described in this document are undoubtedly the *New York Diptych*. The diptych was possibly purchased by De Guzmán between 1637 and 1641 during his viceregal tenure for Philip IV of Spain; De Guzmán brought the panels to Madrid after his return there in October 1643 (Jones 2014, pp. 39–40). We also have evidence that sometime around the seventeenth century the two panels were combined within an elaborate frame with pillars and gables, which is known as a tabernacle frame. This type of frame was popular in the southern Alps (and especially in Italy) since the mid-fifteenth century and was often used to enclose devotional paintings or single-panel altarpieces. Although many possibilities have been suggested, scholars have been unable to determine with full certainty whether the panels of the *New York Diptych* were originally intended to form a diptych, or were wings of a triptych, a sacramental tabernacle, or reliquary (Ainsworth 2017, p. 230). More likely, the *New York Diptych* was used to protect a chest that housed sacred bread or Christ's relics, reminding the congregation of Christ's sacrifice and redemption.

### 2.2. Attribution and Dating

In addition to uncertainty about the provenance and configuration of the *New York Diptych*, scholars have disagreed as to its attribution and dating, stemming largely from its subject matter and style (Belting and Eichberger 1983, pp. 113–43). At one time, art historians proposed that the two panels represented one of the earliest paintings by Jan van Eyck (and his workshop), perhaps created in the 1430s; more recent evidence, however, dates the work to 1440–1441 and does corroborate the van Eyck attribution (Ainsworth 2016, pp. 124–27). Most paintings by Van Eyck—at least those left to us—are "Marian paintings" that presented devotional images; indeed, the *New York Diptych* illustrates narrative scenes focusing on the highlights of Christ's life. Several paintings that show biblical stories have been associated with Van Eyck's workshop, but some remain controversial and thus may not be reliable in providing clues for the diptych's attribution and dating. For example, the *Three Marys at the Sepulcher* (Figure 9) depicts Jerusalem in the background, which is comparable with the townscape represented in the Crucifixion panel of the diptych. Generally, however, the architectural motifs illustrated in the *Three Marys at the Sepulcher* are not persuasive in crediting the work to Van Eyck; similarly, the limbs of the angel and women lack anatomical accuracy, which makes it difficult to attribute the painting with

sufficient certainty to Jan van Eyck. In contrast, the *Crucifixion* (Figure 10) shows more detailed anatomical features on Christ's face and body, which suggest the same or similar hands who made the Crucifixion of the *New York Diptych*.

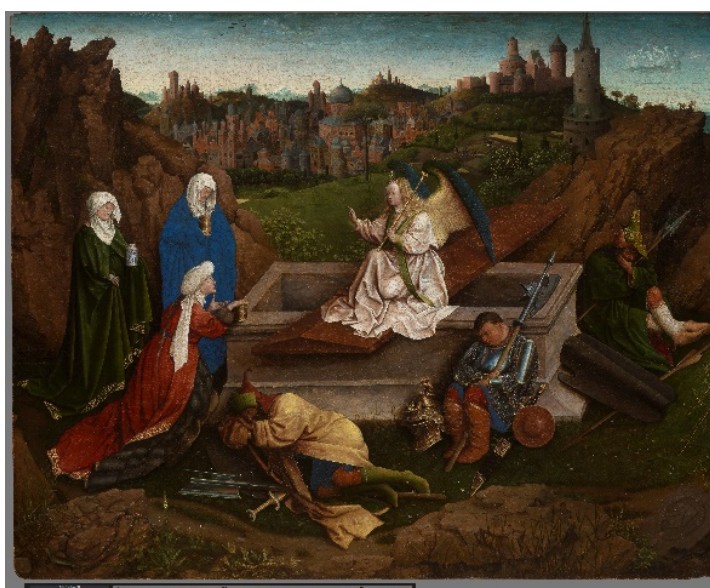

**Figure 9.** Jan van Eyck, workshop, *Three Marys at the Sepulcher*, ca. 1440, Rotterdam, Museum van Boijmans Beuningen.

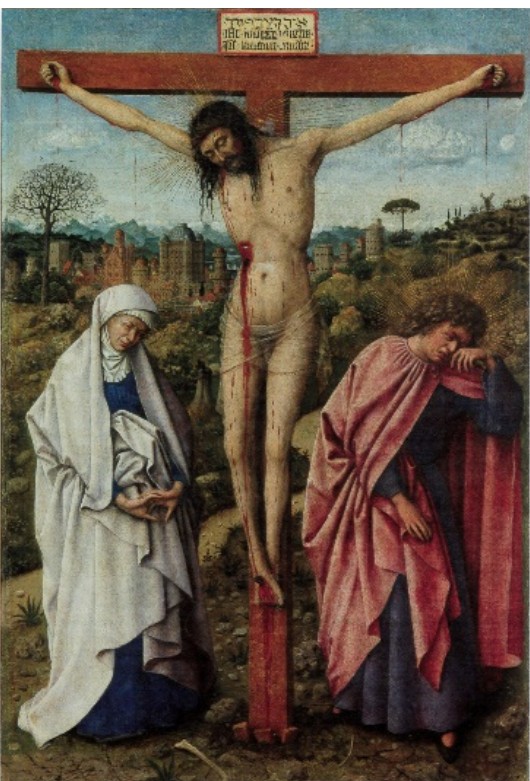

**Figure 10.** Jan van Eyck, workshop, *Crucifixion*, ca. 1430–1450, Berlin, Staatliche Museen zu Berlin.

A clearer relationship with the Crucifixion of the *New York Diptych* can be found in a drawing created by an anonymous artist who used goldpoint and silverpoint to depict the scene (Figure 11). This drawing mirrors the techniques Van Eyck used for his drawing of the *Portrait of Niccolo Albergati* (Camp 2016, pp. 53–54). Although several soldiers and

noblemen represented in the New York Crucifixion appear in the Rotterdam drawing, they are either drawn at different places or in reverse; on this basis, the Rotterdam drawing is generally considered as a pastiche of the New York Crucifixion (Borchert 2016, p. 97).

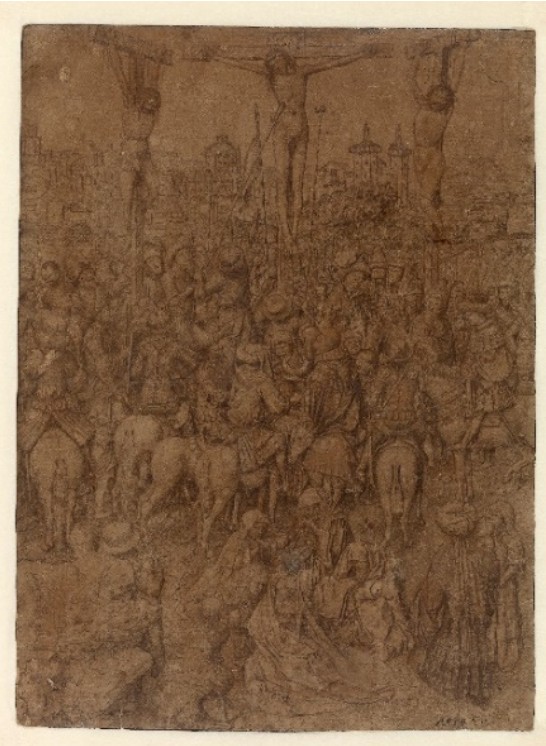

**Figure 11.** Jan van Eyck, workshop or follower, Crucifixion, ca. 1440–1480, Rotterdam, Museum van Boijmans Beuningen.

From the viewpoint of its microscopic representation, the *New York Diptych* can also be compared with several illuminations in the *Turin-Milan Book of Hours*, which is one of the most problematic works associated with Jan van Eyck (Vanwijnsberghe 2020, pp. 296–315). While the attribution, dating, and commissioner of the *Book of Hours* is certainly intriguing, it is beyond the scope of this article. By and large, however, most art historians agree that the Hand G who created the five most remarkable illuminations is likely Jan van Eyck. For instance, a common thread found in these five illuminations is the mature technique deftly employed to represent natural phenomena and stimulate the senses of the audience (Figure 12). Both in the illuminations by Hand G and in the *New York Diptych*, we feel the movement of air, the play of light, and the sounds of nature. Similar to the diptych, a precise dating of the illuminations by Hand G in the *Turin-Milan Book of Hours* remains elusive. Some consider it to be one of the earliest works attributed to Van Eyck, while others place the work as late as the 1440s. Either way, the optical knowledge and representative techniques needed to convey movement in these works of art are sufficiently similar to support their shared patrimony.

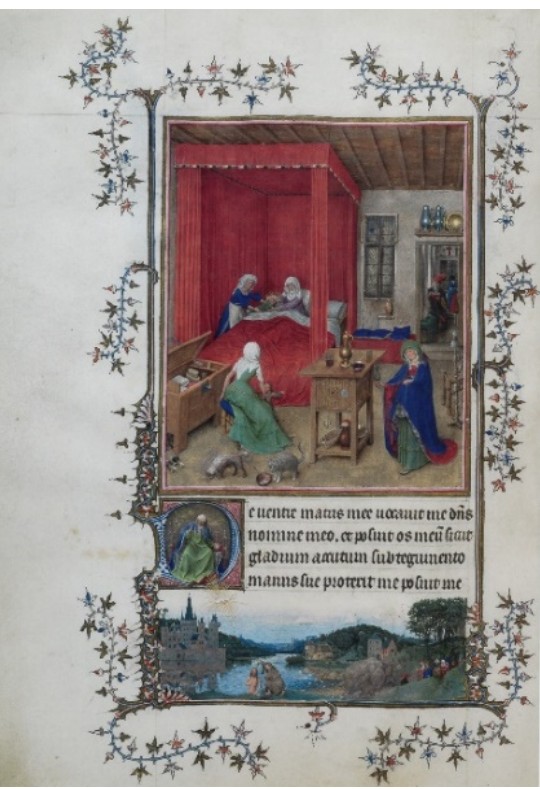

**Figure 12.** Hand G, Birth of Sint John the Baptist, *Turin-Milan Book of Hours*, fol. 93v, ca. 1435–1445 (?), Turin, Biblioteca Nazionale Reale.

### 3. Motif of the Skeleton in the Last Judgment

While the influence of the New York Crucifixion can be seen in several later works associated with Van Eyck's workshop, an unequivocal reference to the Last Judgment panel of the *New York Diptych* can be observed in the two-panel diptych created by Petrus Christus in 1452: *The Annunciation and Nativity and Last Judgment* (also known as the *Berlin Diptych*; Figure 13). Bernhard Ridderbos maintains that the *Berlin Diptych* was originally wings of a triptych, adding that on the reverse sides could be found Saints Peter and Paul in grisaille (Ridderbos 2005, pp. 78–86). While the composition is based on Van Eyck's Last Judgment, Christus, who was a successor of Van Eyck in Bruges, modified the original complex composition and motifs. It is plausible that Christus did not see the original work, but referred to a drawing of the New York Last Judgment in Van Eyck's workshop.

In Van Eyck's Last Judgment, Christ as judge and heavenly ruler, the largest figure in the panel, is positioned at the top and is surrounded by dozens of angels and flanked by large images of the Virgin Mary and John the Baptist. The analogous depiction in Christus's work is somewhat weakened, where we see only four angels and must search for the Virgin Mary and John among many other saints. The land and ocean are represented not on the left and right sides, but in the foreground and background; moreover, the landscape seen in the *Berlin Diptych* is a far less horrific sight, lacking earthly fires and an ocean giving up its lost souls. Similar to Van Eyck's depiction, however, flames leap from a comparable image of the skeleton to burn the cursed souls in hell. In addition, a simplified depiction of hell and its souls is presided over by the pivotal figure of the Archangel Michael wearing simple armor and bearing a sword to subdue the demons at his feet. Here, Christus chooses a dark motif for Michael, and the whole composition is far less complex compared to the New York Last Judgment.

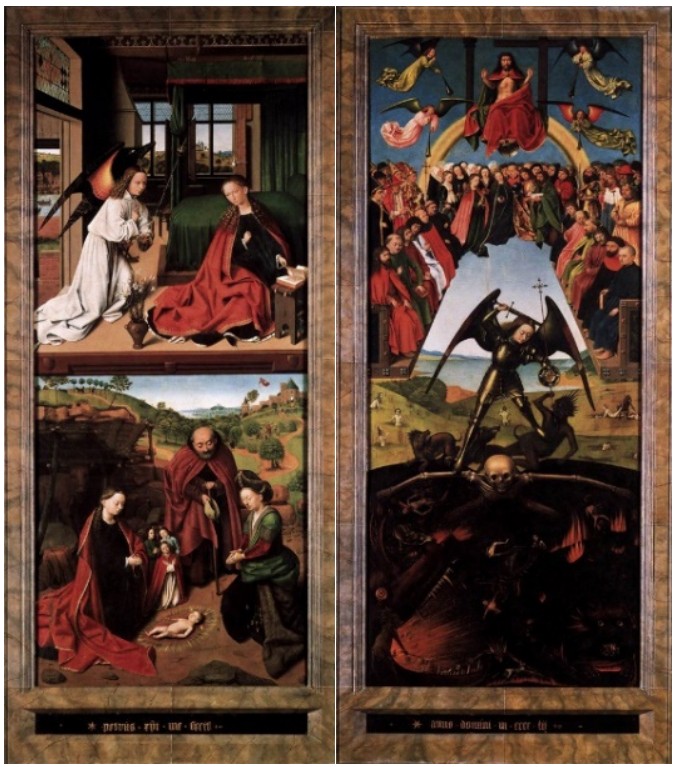

**Figure 13.** Petrus Christus, *Annunciation and Nativity and Last Judgment,* 1452, Berlin, Gemäldegalerie.

In Van Eyck's Last Judgment, infrared reflectography reveals that "MORO" or "MORS" is written on the forehead of the centralized image of the skeleton (Figure 14), on the basis of which Eichberger argues that the skeleton can be considered as "a personification of the second, or the forever death" (Eichberger 1987, p. 87). As a motif of death, a skeleton, or simply a skull, is sometimes represented on outstretched wings in early Netherlandish diptychs or triptychs, serving as a *memento mori* image reminding the viewer that death is inevitable with potentially dreadful consequences. In Hans Memling's *Earthly Vanitas and Divine Salvation* (c. 1480s; Figure 15), separate images of a skeleton and skull are represented with four others: Christ in majesty, a coat of arms, a ghastly depiction of hell and a devil, and one showing a naked young lady admiring her beauty in a mirror, which indicates the damnable sin of vanity or perhaps lust. In the panel showing the dead body we read "Ecce finis hominis comparatus sum luto assimilates sum faville et cineri (Behold the end of man. I am like clay and have becomes as dust and ashes)"; in the panel of the skull is written "SCIO ENIM QVOD REDEMPTOR MEVS VIVIT ET IN NOVISSIMO DIEDETERRA SVRRECTVRVS SIM RVRSVM CIRCV[m]DABOR PELLE MEA ET INCARNE MEA VIDEBO DEV[m] SALVATOREM MEVM IOB XIX CAP (For I know that my redeemer liveth, and that he shall stand at the latter day upon the earth. And though after my skin worms destroy this body, yet in my flesh shall I see God) [Book of Job 19: 25–26]". In the panel representing the monster-like dancing devil, Memling has included the phrase "IN INFERNO NVLLA EST REDEMPTIO (There is no redemption in hell)". By illustrating the relationships between earthly sin, death, and an inescapably hellish fate for the wicked, this series of the panel paintings is intended to remind the audience about the fleeting nature of life and the importance of virtue.

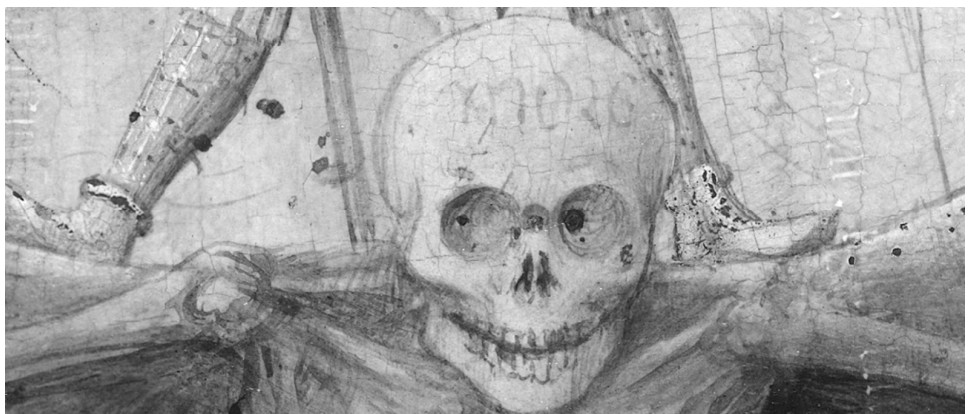

**Figure 14.** Infrared reflectography of the skeleton, detail of Figure 1.

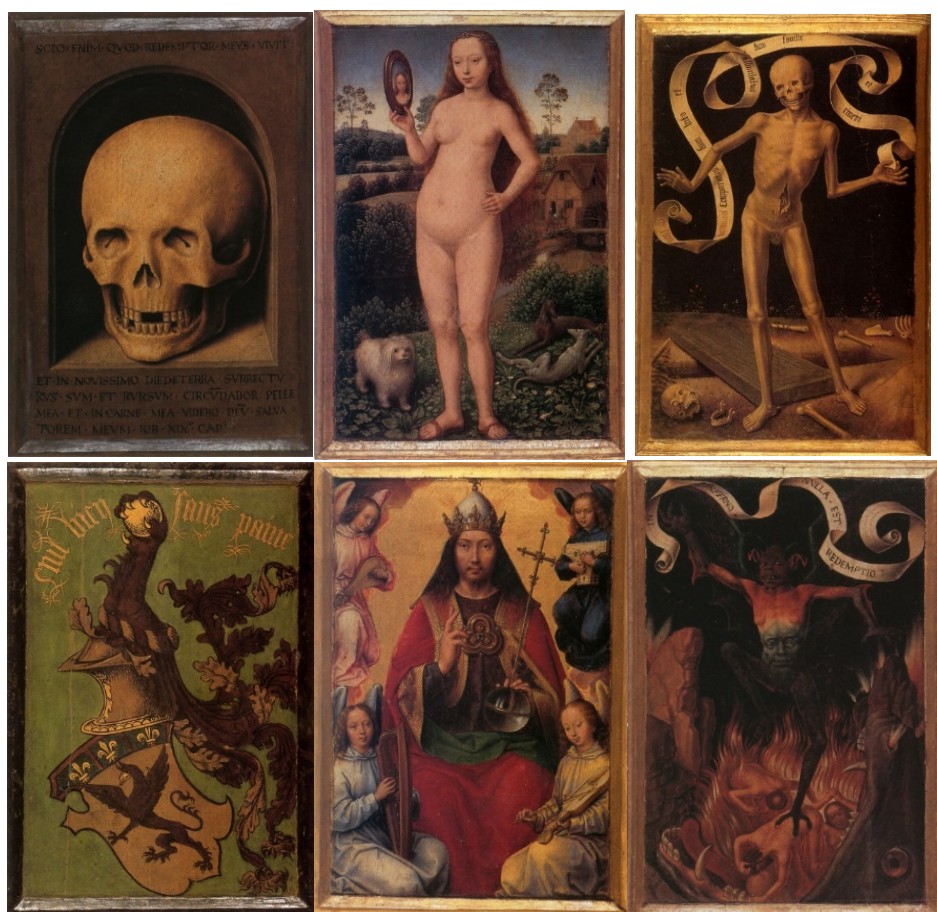

**Figure 15.** Hans Memling, *Earthly Vanity and Divine Salvation*, ca. 1480s, Strasbourg, Musée des Beaux-Arts.

While its iconographical source is unknown, the skeleton in the New York Last Judgment may have served as a similar *memento mori* image, while also functioning as a physical border between heaven and the earthly realms in the center of the panel, and the dark and gruesome reality of hell depicted in the bottom half. Whether Van Eyck intended it or not, this lower depiction attracts our eyes more than Christ in majesty or the Archangel Michael—not only because the skeleton is placed in the middle of the Last Judgment, but also because it gazes menacingly and directly at the viewer. Stretching out its boney limbs and voluminous wings, the lively skeleton reminds us of corpses represented in the *dance macabre*, which was first represented in the Holy Innocents Cemetery in Paris in 1424 and

became popular in the fifteenth and sixteenth centuries. Like those in the *dance macabre*, the skeleton in the New York Last Judgment tells us that death will capture everyone, no matter rich or poor, at any time, and anywhere.

## 4. Possible Source of the Inscriptions

On the skeleton's wings we read "CHAOS MAGNV[M]" and "VMBRA MORTIS". These inscriptions are also written in the underdrawing in slightly different locations. It is worth reconsidering why "CHAOS MAGNV[M]", which is, as mentioned before, referred to only once in the Bible, is written on one of the skeleton's wings. This particular phrase appears in the parable of Jesus and Lazarus (Luke 16:26). There was a rich man who lived in luxury, and at his gate lay a beggar named Lazarus. When Lazarus died, the angels carried him to Abraham's side. When the rich man died and found himself tormented in Hades, he looked up and saw Abraham far away with Lazarus by his side. Abraham said to the rich man, "Between us and you a great chasm has been set in place, so that those who want to go from here to you cannot, nor can anyone cross over from there to us". The place where Lazarus' soul found rest is known as the Bosom of Abraham (Figure 16). The parable of the rich man and Lazarus has often been cited in theological discussions of where the blessed and cursed are destined to remain until the Last Judgment. For instance, Tertullian (d. after 220) referred to the Bosom of Abraham as *interim refrigerium* and to the place for the rich man as *interim tormentum* (Mecklenburg and Mertens 2013, p. xix).

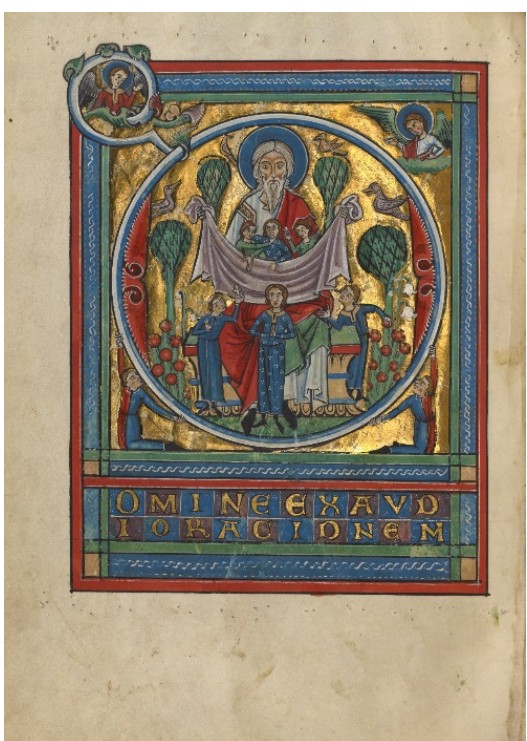

**Figure 16.** Initial D: Souls in the Bosom of Abraham, ca. 1240–1250, Ms. Ludwig VIII 2, fol. 113v, Los Angeles, The J. Paul Getty Museum.

But why is "CHAOS MAGNV[M]" combined with "VMBRA MORTIS" on the skeleton's wings in Van Eyck's Last Judgment? Is there any context for linking the two terms in this painting? Although this question has never been considered, the current author suggests that the texts on Death's wings are cited from the fourth sermon delivered by Bernard of Clairvaux on All Saints' Day (See Appendix A). In this sermon, Saint Bernard compares the Bosom of Abraham with the lower part of the altar where John the Evangelist heard the voices of saints. Referring to the places for holy souls and for those of the wicked, he states,

Sed iam venerat, fratres, tempus illud quod beatus Iob postulabat, iam recordandi tempus, iam venerat tempus miserendi, quando Sanctorum voces sub altare Dei beatus Ioannes audivit. . . . Donec enim veniret desideratus ille, qui sanguine suo deleret chirographum damnationis nostrae et, flammeum exstinguens gladium, aperiret credentibus regna caelorum, nullus omnino cuiquam Sanctorum ad ea patebat accessus; sed providerat eis Dominus in inferno ipso locum quietis et refrigerii, chaos magnum firmans inter sanctas illas animas et animas impriorum. Quamvis enim utraeque in tenebris essent, non utraeque erant in poenis; sed cruciabantur impii, iusti vero consolabantur. . . . Hunc ergo locum, obscurum quidem, sed quietum, sinum Abrahae Dominus vocat. . . . In hunc ergo locum Salvator descendens, CONTRIVIT PORTAS AEREAS ET VECTES FERREOS CONFREGIT, eductosque vinctos de domo carceris, sedentes quidem, hoc est quies centes sed in tenebris et umbra mortis, iam tunc quidem sub altare Dei collocavit (But when Blessed John heard the voices of the saints beneath the altar, the time for which blessed Job asked had already come; the time of reckoning, the time of mercy had already come. . . . For until he who was longed for came to cancel the decree of our damnation with his own blood and quench the fiery sword, opening the kingdom of heaven to those who believe, no way to it lay open to any of the saints. Rather, God had provided a place of rest and refuge for them below, and a great chasm was fixed between those holy souls and the souls of the wicked. For although both were in darkness, they were not both undergoing punishment; the wicked were in torment and the righteous in consultation. . . . And this place, dark indeed but restful, the Lord called the bosom of Abraham because, I suppose they rested in faith and expectation of the Savior. . . . Now the Savior, descending to this place, broke the gates of brass and shattered the bars of iron and led from their prison those who had been bound; they were indeed seated, that is, at rest, but in the shadow of death. Then he set them beneath the altar of God.) (Leclercq and Rochais 1968, pp. 354–55; Scott 2016, pp. 238–40, with some modifications of the translation by the current author).

Bernard indicates that the Bosom of Abraham was a dark place where both the holy souls awaiting judgment rested in faith and expectation of the Savior's mercy, while the souls of the wicked were locked in torment in the underworld. In the end, the Lord would come to free the righteous and leave the wicked to their fate. If the inscriptions on the skeleton's wings are cited from this sermon, they can be interpreted in two ways. First, from a visual perspective, the stretched wings represent the enormity of the chasm that divides heaven and hell, while also casting the shadow of death over the underworld. From this viewpoint, the skeleton's wings contrast starkly with the voluminous blue mantle of the Merciful Virgin, which protects the blessed souls in heaven. If we consider the inscriptions to indicate the realm of underworld, the skeleton can be seen as a kind of meta image: an antipole of the Bosom of Abraham. In other words, the two phrases, when combined with the image of Death, remind the viewer that the place for sinful souls is divided from that of the blessed by a great chasm. Thus, the New York Last Judgment is not a traditional representation of the Last Judgment reminding the viewer of what awaits them upon death and until the Last Judgment.

It must be stressed, however, that it cannot be concluded with complete certainty that the inscriptions were directly cited from the sermon itself. Nonetheless, given that the centuries-earlier writings of Bernard were frequently cited in later theological literature and several works of art, we can surmise that they were still very influential in the fifteenth century when Van Eyck created the Last Judgment. His influence can be found not only in numerous mentions, but also in representations, of his miraculous vision in which the Virgin ejects her milk into his mouth. In the *Crucifixion* attributed to Rogier van der Weyden (ca. 1438–1440, Berlin, Staatliche Museen zu Berlin—Preussischer Kulturbesitz, Gemäldegalerie), for example, we read the inscriptions emerging from the mouth of the Virgin to her son "O fili[us] dignare me attrahere et crucis I pedem manus figere. Bernhardus

(Oh son, let me draw close and take the foot of the cross in my hands. Bernard)". While this Marian lamentation is now attributed to Cistercian Oglerius (1136–1214), it was at that time attributed to Bernard (Bohde 2019, pp. 39–40). If the texts in the Last Judgment of the *New York Diptych* were cited, directly or indirectly, from Bernard's sermon, it is possible that a theological adviser who had rich knowledge of his teachings may have suggested the theme to the person who commissioned the work. The inscriptions on the figure of Death's wing humbly indicate that the *New York Diptych* emerged from a rich theological environment and that it was intended for the learned audience who could connect the texts with Bernard's sermon—or one could assume that some kind of sermon was delivered in front of the image to edify the audience.

## 5. Conclusions

This short paper discusses the influences of the *New York Diptych* and, for the first time, suggested a possible source for the two inscriptions written on the wings of the skeleton depicted in Van Eyck's Last Judgment, which to date have been considered simply as a metaphor for death and the underworld. In this author's estimation, the iconography of the Last Judgment, which shows the moment when all souls will be judged by God, fits well with the theme of Saint Bernard's sermon for All Saints' Day. As Alfred Acres demonstrates concerning Rogier van der Weyden's works, "painted texts" is an impulse to cultivate a role of the audience in the creation of meaning (Acres 2000). They also provide a key to consider the commissioner, intended audience, and a milieu of the creation. It is time to consider the *New York Diptych* from this new perspective.

As for relationships between medieval sermons and other works by Van Eyck, we know that the discussion by Honorius Augustodunensis in the introduction to his *Sigillum Beatae Mariae Virginis* contains imagery pertinent to the *Virgin of Nicolas Rolin* (Van Buren 1978), while the writings of Rupert of Deutz may have been referenced to make the iconographical plan for the *Ghent Altarpiece* (Dhanens 1973). Both Nicolas Rolin and Jodocus Vijd, as well as Van Eyck himself, were likely to have engaged with theological advisers in their connection with the highly intellectual circle of the Burgundian court to which they belonged or frequented. Accordingly, one could assume that the individual who commissioned or gave advice on the *New York Diptych* could have had knowledge of Bernard's sermon and suggested the inclusion of a reference to it. One may also assume that, if the commissioner was integrated in the network of the Burgundian court, like Rolin and Vijd, he could also have had access to theological literature kept in the rich library of Philip the Good, Duke of Burgundy, for whom Van Eyck served as a court painter. The library covered all fields of medieval thought, including theology and religious philosophy, and many manuscripts were transcribed at the express request of the Duke by copyists (Bousmanne and Savini 2020). One may further trace the relationship between Bernard's writing, Van Eyck's painting, and possibly Philip the Good in another painting by Jan van Eyck. As pointed out by Millard Meiss, various Annunciation scenes produced in the fifteenth century include the motif of the light through the glass, which reflects the famous allusion of the light and unbroken window attributed to Bernard (Meiss 1945):

> Just as the brilliance of the sun fills and penetrates a glass window without damaging it, and pierces its solid form with imperceptible subtlety, neither hurting it when entering nor destroying it when emerging: thus the word of God, the splendor of the Father, entering the virgin chamber and then came forth from the closed womb. (Meiss 1945, p. 176)

Among the paintings including the motif of the rays through the glass is the *Washington Annunciation* by Jan van Eyck (ca. 1434/36) (Meiss 1945, p. 178), which is mentioned as coming from Dijon and painted for Philip the Good (Hand and Wolff 1986, p. 81). Are both the *New York Diptych* and *Annunciation* reflecting Philip's affinity with Bernard's writing? Further research on circulation of Bernard's writings in and around the ducal court will provide a clue for the identity of the commissioner.

**Funding:** This research received no external funding.

**Institutional Review Board Statement:** Not applicable.

**Informed Consent Statement:** Not applicable.

**Conflicts of Interest:** The author declares no conflict of interest.

## Appendix A

Fourth Sermon for All Saints' Day by Bernard of Clairvaux.

De sinu Abrahae et al.tari sub quo Sanctorum animas Beatus Ioannes audivit, et septem panibus ex quibus totidem sportae remansisse leguntur.

Cum de altari illo caelesti, sub quo beatus Ioannes Sanctorum voces audivit, sermo incidisset, distulimus, sincut meminisse arbitror caritatem vestram, ut, oratione praemissa, securior nobis ad tam sacrum secretumque cubiculum pateret accessus. Tempus est ut dicamus iam quod de eo nobis sentire datum est, sine praeiudicio sane, si cui forte aliter fuerit revelatum. Illud ergo primum movere potest, quid sibi velit quod beatus Ioannes sub altare Dei audisse se perhibet sanctarum voces animarum, cum Salvator in Evangelio de Lazari anima loquens, non sub altare Dei, sed in sinum Abrahae dicat eam ab angelis deportatam. Nam et sanctus Iob, ut apparet, nequaquam ad altare Dei ausus est aspirare, cum diceret: QUIS MIHI TRIBUAT UT IN INFERNO PROTEGAS ME ET ABSCONDAS ME, DONEC PERTRANSEAT FUROR TUUS ET CONSTITUAS MIHI TEMPUS, IN QUO RECORDERIS MEI? Sed iam venerat, fratres, tempus illud quod beatus Iob postulabat, iam recordandi tempus, iam venerat tempus miserendi, quando Sanctorum voces sub altare Dei beatus Ioannes audivit. Donec enim veniret desideratus ille, qui sanguine suo deleret chirographum damnationis nostrae et, flammeum exstinguens gladium, aperiret credentibus regna caelorum, nullus omnino cuiquam Sanctorum ad ea patebat accessus; sed providerat eis Dominus in inferno ipso locum quietis et refrigerii, chaos magnum firmans inter sanctas illas animas et animas impriorum. Quamvis enim utraeque in tenebris essent, non utraeque erant in poenis; sed cruciabantur impii, iusti vero consolabantur. Quod autem in tenebris essent, beati Iob testimonio didicimus, qui se quoque in locum tenebrosum et opertum mortis caligine perhibebat iturum. Hunc ergo locum, obscurum quidem, sed quietum, sinum Abrahae Dominus vocat; pro eo, ut arbitror, quod in fide et exspectatione quiescerent Salvatoris. Abrahae enim fides tam manifeste probata est, et approbata, ut primus ipse futurae incarnationis Christi meruisset accipere promissionem. In hunc ergo locum Salvator descendens, CONTRIVIT PORTAS AEREAS ET VECTES FERREOS CONFREGIT, eductosque vinctos de domo carceris, sedentes quidem, hoc est quies centes sed in tenebris et umbra mortis, iam tunc quidem sub altare Dei collocavit, abscondens eos in tabernaculo suo in die malorum et protegens eos in abscondito tabernaculi sui, donec veniat tempus quo procedant, completo iam numero fratrum, et percipiant regnum quod eis paratum est ab origine mundi. Iam vero sicubi forte praesens quoque Sanctorum requies Abrahae sinus vocatur, certum est de Evangelio hanc inolevisse consuetudinem, licet neminem oporteat dubitare longe alium hunc sinum esse quam illum, quippe cum ille in tenebris, hic in luce multa, in inferno ille fuerit, iste in caelo. Non incongrue tamen etiam nunc dictum videtur Patriarcharum filios paternum in sinum recipi, cum ad eorum consortium ab hoc saeculo transire meruerint. (Leclercq and Rochais 1968, pp. 354–55).

## Note

[1]  A comprehensive bibliography of the *New York Diptych* can be found in the Metropolitan Museum of Art website: https://www.metmuseum.org/art/collection/search/436282 (accessed on 16 December 2021).

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
