# Peer review of "Jan van Eyck’s New York Diptych: A New Reading on the Skeleton of the Great Chasm"

_arts, 1998_

Round 1

Reviewer 1 Report

It is this reviewer's hope that the author will not think that the average score above does not mean this can go into print. The article brings a new notion into scholarship and that is all very fine. The article would read better if the possible patron(s) and provenance were introduced at an earlier point. The article would also gain importance if the dissemination of Bernard's sermon was actually known in painter's and patron's circles at the time, i.e. include more context and sources to lines 386-388. Moreover, an engagement with Alfred Acres' article "Rogier van der Weyden's painted texts" Artibus et historiae 41 (2000) would situate this reading more fully into the  painter/patrons/religious practices at the time.
See added pdf for more specific comments. 

Reviewer 2 Report

I made some extensive comments through the initial sections relating to the papers structure, that I think can be cleaned up more.  I would suggest make other connections between Clairrvaux's influence on van Eyck-I'm thinking of some of the Madonna images and floral gardens to round out the connection to the All Saints suggestion.  

In general though see comments throughout the text to assist in improving this essay.

I also provide an important suggestion for providing more clarity on the text in question. I'll paste that comment here:

"Aside from my general comments on the structure, I would strongly suggest the author download the high res image from the MET and run the skeleton through the app: Remini that can be used on iphone or android based phones. The filter will fine tune the image, to see the text clearly.  Once I did this and flipped the image on both its horizontal and vertical axis, the letters became really clear and confirming the author’s and Eichberger’s thoughts of what the letter depicts. The author may not need to do the rotations, because it does cause the letters to appear backwards, but it made the letters more legible for me on my Macbook."
